# Effects of Maternal Gut Microbiota-Targeted Therapy on the Programming of Nonalcoholic Fatty Liver Disease in Dams and Fetuses, Related to a Prenatal High-Fat Diet

**DOI:** 10.3390/nu14194004

**Published:** 2022-09-27

**Authors:** Hong-Ren Yu, Jiunn-Ming Sheen, Chih-Yao Hou, I-Chun Lin, Li-Tung Huang, You-Lin Tain, Hsin-Hsin Cheng, Yun-Ju Lai, Yu-Ju Lin, Mao-Meng Tiao, Ching-Chou Tsai

**Affiliations:** 1Department of Pediatrics, Chang Gung Memorial Hospital-Kaohsiung Medical Center, Graduate Institute of Clinical Medical Science, Chang Gung University College of Medicine, Kaohsiung 833, Taiwan; 2Department of Pediatrics, Chiayi Chang Gung Memorial Hospital, Puzi City 613, Taiwan; 3Department of Seafood Science, National Kaohsiung University of Science and Technology, Kaohsiung 811, Taiwan; 4Department of Obstetrics and Gynecology, Chang Gung Memorial Hospital-Kaohsiung Medical Center, Kaohsiung 833, Taiwan

**Keywords:** prenatal, high-fat diet, DOHaD, nonalcoholic fatty liver disease, *Lactobacillus reuteri*, butyrate

## Abstract

Metabolic disorders can start in utero. Maternal transmission of metabolic phenotypes may increase the risks of adverse metabolic outcomes, such as nonalcoholic fatty liver disease (NAFLD); effective intervention is essential to prevent this. The gut microbiome plays a crucial role in fat storage, energy metabolism, and NAFLD. We investigated the therapeutic use of probiotic *Lactobacillus reuteri* and postbiotic butyrate gestation in the prevention of perinatal high-fat diet-induced programmed hepatic steatosis in the offspring of pregnant Sprague–Dawley rats who received regular chow or a high-fat (HF) diet 8 weeks before mating. *L. reuteri* or sodium butyrate was administered via oral gavage to the gestated rats until their sacrifice on day 21 of gestation. Both treatments improved liver steatosis in pregnant dams; *L. reuteri* had a superior effect. *L. reuteri* ameliorated obesity and altered the metabolic profiles of obese gravid dams. Maternal *L. reuteri* therapy prevented maternal HF diet-induced fetal liver steatosis, and reformed placental remodeling and oxidative injury. Probiotic therapy can restore lipid dysmetabolism in the fetal liver, modulate nutrient-sensing molecules in the placenta, and mediate the short-chain fatty acid signaling cascade. The therapeutic effects of maternal *L. reuteri* on maternal NAFLD and NAFLD reprogramming in offspring should be validated for further clinical translation.

## 1. Introduction

Evidence indicates that maternal obesity will be a major determinant of the “developmental origins of health and disease” in the next generation. Obesity before pregnancy and gestational weight gain, especially in early pregnancy, may increase the risk of obesity in the offspring [1,2,3]. Adverse cardiovascular risks in the offspring, such as coronary heart disease, type 2 diabetes, and stroke, also increase from childhood to adulthood [1,4].

Non-alcoholic fatty liver disease (NAFLD), an excessive accumulation of fat in the liver, is the most common form of chronic liver disease and is closely related to obesity. A meta-analysis conducted in Asia between 1999 and 2019 showed that the incidence of NAFLD was 50.9 cases per 1000 person-years [5]. Yunosi et al. estimated the global prevalence of NAFLD to be as high as 25% [6]. In an autopsy study conducted in San Diego, the authors enrolled 742 children and found that the prevalence of fatty liver was 9.6%, whereas the prevalence in obese children was as high as 38% [7]. NAFLD is strongly associated with metabolic syndrome and insulin resistance. NAFLD can eventually progress to hepatic cirrhosis, cardiovascular problems, malignancy, and chronic kidney disease [8,9]. Hepatic steatosis in children can be observed in early childhood developmental stages and even in utero [10]. The prenatal predisposing factors for NAFLD include maternal obesity, maternal metabolic syndrome, and gestational diabetes [11,12]. Fetuses nurtured by dams with high-fat (HF) diets during pregnancy can develop fatty livers [13]. The disease process of NAFLD is currently considered to be related to abnormal lipid metabolism, inflammation, and oxidative stress [14,15]. In a previous animal study of obese dams fed HF diets, we showed that the alteration of the placenta through oxidative stress and metabolism-related transcriptomes and a change in the short-chain free fatty acid cascade leads to lipid dysmetabolism and steatotic changes in the fetal liver. Pregnant women are more likely to be on HF diets and tend to be obese; therefore, providing a safe and effective strategy to prevent fatty liver in the offspring during pregnancy is an important issue.

Many studies have shown an association between gut dysbiosis and metabolic syndrome, obesity, type 2 diabetes, and NAFLD [16,17,18,19]. Ge et al. showed that adult patients with NAFLD demonstrated lower diversity and a *Firmicutes/Bacteroides* (F/B) ratio than the controls [20]. Chierico et al. found an increased abundance of *Bradyrhizobium, Anaerococcus, Peptoniphilus, Propionibacterium acnes, Dorea*, and *Ruminococcus*, and a decreased abundance of *Rikenellaceae* and *Oscillospira* in the gut microbiome of children with NAFLD [16]. In animal studies, the gut microbiota of pregnant dams with hepatic steatosis correlating with a HF diet showed a low level of alpha diversity and *Lachnospiraceae* genus, but more *Romboutsia* and *Akkermansia* genera than the control group [13]. Current strategies to alleviate NAFLD primarily include lifestyle changes, such as exercise, weight loss, and dietary control. Since the gut microbiome plays a crucial role in body fat storage, energy metabolism, and inflammatory response, these mechanisms also correspond to the fatty liver. Manipulation of the gut microbiota has the potential to be a deprogramming strategy for NAFLD in pregnant mothers and even offspring.

The interaction between the gut microbiota and the host is mediated via its metabolites, which are produced by the fermentation of food substances by the gut microbiota [21]. Postbiotics are substances released or produced through the metabolic activities of microorganisms that exert beneficial effects on the host [22]. Short-chain fatty acids (SCFAs) are polysaccharide fermentation products produced by the gut microbiota. SCFAs have been shown to modulate glucose homeostasis, lipid metabolism, and immunity [23]. SCFAs have also been shown to improve the gut barrier and mucosal immune function, and ameliorate a HF diet-induced fatty liver [24,25,26,27]. The most well-known products of SCFAs are acetic acid, propionic acid, and butyric acid. Among these, butyrate acids have received the most attention. Butyrate is the primary energy source for colonocytes. However, it is uncertain whether treatment with butyrate can ameliorate hepatic abnormalities generated by diet-induced maternal obesity.

Observations on the long-term effects of maternal obesity on offspring have significant public health implications. The prevalence of obesity in women of reproductive age is increasing worldwide, thus contributing to an increased risk of exposure of the offspring to an “obese intrauterine environment.” The latter environment perpetuates the vicious cycle of maternal–offspring obesity and increases the burden of chronic diseases. There is an urgent need to develop safe and effective interventions to stop this vicious cycle. In this study, we examined the use of *Lactobacillus reuteri (L. reuteri)* and sodium butyrate to attenuate the effects of maternal obesity on the offspring of pregnant dams.

## 2. Material and Methods

### 2.1. Study Animals and Experimental Design

Seven-week-old virgin female Sprague–Dawley (SD) rats purchased from BioLASCO (Taipei, Taiwan) were housed in a humidity-, temperature-, and light-controlled environment [13]. Food and sterile tap water were provided ad libitum. At one week of adaptation to the experimental environment, the rats were weight-matched and assigned to receive either a HF diet or a regular control diet. The Institutional Animal Care and Use Committee of the Chang Gung Memorial Hospital approved the experimental protocol (no. 2019053001).

The HF diet (D12331) and the control diet feeds were purchased from the Research Diets (Research Diets, New Brunswick, NJ, USA) and Fwusow Industry (Fwusow Industry, Taichung, Taiwan), respectively. The ingredients of the dietary feeds are listed in Appendix A. The rats were fed the specified diet for eight weeks and then mated for two days.

Dams and their fetuses were divided into four groups randomly: control chow (CC), high-fat (HF), high-fat and *L. reuteri* (H + L), and high-fat and sodium butyrate H + B (n = 6 for each group). Two pregnant dams were assigned to each group, and each group produced approximately 20 offspring. “Group I CC”: maternal rats commenced a control chow diet prior to mating, and were continued on this diet until their sacrifice on gestational day 21 (GD21). “Group II HF”: maternal rats commenced a HF diet 8 weeks prior to mating and were fed this diet until their sacrifice on GD21. “Group III H + L”: maternal rats commenced a HF diet 8 weeks prior to mating and the pregnant rats were administered *L. reuteri* (GMNL-89, GenMont Biotech, Inc., Taipei, Taiwan) (1 × 10^9^ colony-forming units (CFU)/day) by gavage from GD0 until their sacrifice on GD21. “Group IV H + B”: maternal rats commenced a HF diet 8 weeks prior to mating and the pregnant rats were administered 1% (*w*/*v*) sodium butyrate (B5887, SIGMA, St. Louis, MO, USA) in drinking water (150 mg/kg/day) by gavage from GD0 until their sacrifice.

For further experiments on *L. reuteri*, other groups were included. “Group I CC”: maternal rats were foddered a control chow diet before mating and during gestation until sacrifice on GD21. “Group II HF”: maternal rats were foddered a HF diet for 8 weeks before mating and during gestation until sacrifice on GD21. “Group III L”: maternal rats were foddered a control chow diet for 8 weeks before mating and during gestation, and the pregnant rats were administered *L. reuteri* (1 × 10^9^ CFU/day) from GD0 until GD21. “Group IV H + L”: maternal rats were foddered a HF diet for 8 weeks before mating and during gestation, and the pregnant dams were administered *L. reuteri* (1 × 10^9^ CFU/day) from GD0 until sacrifice on GD21.

### 2.2. Specimen Collection

The dams were sacrificed by anesthetization with Zoletil (25 mg/kg) (tiletamine-zolazepam, Virbac; Carros Cedex, France) and Rompun (23.32 mg xylazine hydrochloride, Bayer, Korea) in a 1:1 mixture by intramuscular injection, followed by cardiac puncture and perfusion. Heparinized blood samples were obtained via a cardiac puncture [13,28]. After the dams were sacrificed, the placenta, liver tissue, and fetal liver were collected by cesarean section as previously described [13]. A portion of the tissue was fixed in 10% formalin for histological analysis, and the remainder was stored at −80 °C for quantitative reverse transcription PCR (RT-qPCR) study.

### 2.3. Blood Pressure (BP) and Body Weight (BW) Measuring

The dams were subjected to weekly BW checks. The indirect tail-cuff method (BP-2000, Visitech Systems, Apex, NC, USA) was used for BP measurements five days before sacrifice.

### 2.4. Intraperitoneal Glucose Tolerance Test

For the intraperitoneal glucose tolerance test (IPGTT), the rats were fasted for 8 h, and 50% glucose (4 mL/kg of BW) was injected intraperitoneally. The tail vein was used to determine serum sugar levels via a blood glucose meter (Accu-Chek, Roche, Germany) before glucose injection and at 15, 30, 60, and 120 min after glucose injection [13,29]. The integrated area under the curve (AUC) of IPGTT was computed using the trapezoidal method.

### 2.5. Biochemical Analysis

Glutamic-oxaloacetic transaminase (GOT), glutamic-pyruvic transaminase (GPT), and total cholesterol (T-chol) levels were measured using an automatic biochemical analyzer (Hitachi model 7450; Hitachi, Tokyo, Japan), according to the manufacturer’s manuals [30]. An enzyme-linked immunosorbent assay (ELISA) kit (Abcam, Cambridge, MA, USA) was used to determine the serum leptin levels (n = 6 per group). The ELISA assay principle relies on the formation of an antibody/antigen complex that attaches the target to the surface of the detection plate and allows the target to be detected and quantified.

### 2.6. Histological Analysis of the Placenta and Liver Tissue

Placenta and liver tissues were fixed with 4% paraformaldehyde at 4 °C overnight, dehydrated in a gradient of ethanol, hyalinized in xylene, and embedded in paraffin wax. Formalin-fixed tissues were cut and stained with hematoxylin and eosin (H&E). The slides were then scanned using a 3DHISTECH PANNORAMIC.SCAN slide scanner. Lipid accumulation in the liver was quantified using ImageJ software (Fiji version 1.8.0).

Oxidative stress of the tissue was measured using 8-hydroxy-2-deoxyguanosine (8-OHdG), a product of DNA oxidation. Briefly, after transferring to polylysine-coated slides, the tissue sections were stained with an anti-8-OHdG antibody (Santa Cruz Biotechnology, Inc., Santa Cruz, CA, USA for 60 min at room temperature and a secondary antibody for 30 min after rinsing. Avidin and biotinylated horseradish peroxidase H were used to visualize the staining.

### 2.7. RNA Isolation and RT-qPCR Analysis

Total RNA was extracted from the tissues using the TRIzol reagent (Invitrogen, Carlsbad, CA, USA). Messenger RNA (mRNA) expression in the fetal liver and placenta tissue was determined by RT-qPCR, as previously reported [13,31]. mRNA primers are listed in Appendix A. Glyceraldehyde 3-phosphate dehydrogenase (*GAPDH*) and 18S ribosomal RNA (rRNA) were used as the housekeeping genes for the liver and placenta, respectively [13]. The comparative threshold cycle method was used to determine relative mRNA quantification [31].

### 2.8. Microbial Analysis

Fecal samples were collected and stored at −80 °C for 1 week prior to sacrifice. The EZNA Soil DNA Kit (Omega. Bio-tek, Norcross, GA) was used to extract microbial DNA from stool samples. The bacterial 16S rRNA gene was amplified by RT-qPCR as previously described [32]. Sequencing amplicons were quantified using QuantiFluor-ST (Promega, San Luis Obispo, CA, USA) and analyzed on an Illumina MiSeq platform.

Amplicon sequencing was performed using 300 bp paired-end raw reads, and the entire paired-end reads were assembled using FLASH v1.2.11. We carried out de-multiplexing based on barcode identification. For quality control, low-quality reads (Q < 20) were excluded from the QIIME v1.9.1 pipeline. The reads were truncated for three consecutive bases (<Q20). The dataset was retained for more than 75% of the original length using split_libraries_fastq.py script in QIIME. UCHIME was used as a chimera check to obtain effective tags, and it was filtered from the data set before operational taxonomic unit (OTU) clustering at 97% sequence identity using the UPARSE function in the USEARCH v7.0.1090 pipeline. The RDP classifier (v2.2) algorithm was employed to annotate the taxonomic classification for each representative sequence based on information retrieved from the Silva Database v132, and was performed with an 80% minimum confidence threshold to record an assignment. We filtered out sequences with one-time occurrences (or if they were present in only one sample). Multiple sequence alignments were conducted using PyNAST software (v1.2) against the core-set dataset in the Silva Database v132 to investigate the sequence similarities among different OTUs. A phylogenetic tree was constructed with a set of sequences representative of the OTUs using FastTree.

For sequence similarities among different OTUs, multiple sequence alignments were conducted by using the PyNAST software (v1.2) against the core-set dataset in the Silva Database v132. A phylogenetic tree was constructed with a set of sequences representative of the OTUs using FastTree.

To normalize the variations in sequence depths across samples, OTU abundance information was rarefied to the minimum sequence depth using the QIIME script (single_rarefaction.py). Subsequent analyses of alpha and beta diversities were performed using normalized data.

### 2.9. SCFAs Analysis

Gas chromatography was used to measure the plasma concentrations of acetic acid, propionic acid, and butyric acid [32]. Briefly, 5 μL of 100 μM internal standard was mixed with 100 μL of the sample and 100 μL of propyl formate. After vortexing and centrifugation, the supernatant was subjected to gas chromatography (GC) analysis using a Shimadzu QPlus 2010 gas chromatograph.

### 2.10. Statistics

Continuous data were analyzed using the Kruskal–Wallis test or the one-way analysis of variance (ANOVA) with Tukey post hoc tests, as indicated; *p* value < 0.05 was considered statistically significant. Values are expressed as the mean ± standard error of the mean. The BW difference and the IPGTT test data between the groups were determined by repeated measures analysis. The interaction between the group and time (G × T) was calculated for each variable. Statistical analyses were performed using the Statistical Package for the Social Sciences (SPSS) software version 19. We calculated the sample size as four rats per group with the glucose level at a 15 min mean difference between the two groups of 152.87 (=386.20 − 233.33) mg/dL (standard deviation) 50.97), this provided 80% power to detect such a difference using a two-sample *t*-test with a two-sided type I error of 0.05. We recruited 24 rats (six per group) to achieve roughly four rats per treatment group. The power test was performed using the G*Power 3.1.9.4.

## 3. Results

### 3.1. L. reuteri Intervention Ameliorates HF Diet-Related Obesity and Alters the Metabolic Profiles of Dams

Changes in the weekly BW of the animals are illustrated in Figure 1 and listed in Appendix A. After one week of diet manipulation, the dams receiving the HF diet had heavier BW than those receiving the control diet (234.30 ± 6.45 g vs. 210.28 ± 3.06 g; *p* = 0.032). In this study, *L. reuteri* and butyrate were administered after breeding. We found that *L. reuteri* treatment decreased the obesity of dams receiving the HF diet after two weeks of treatment (HF group vs. HF plus *L. reuteri* group; 358.26 ± 12.86 g vs. 325.41 ± 11.61 g; *p* = 0.021). Before delivery, the HF group dams maintained a higher BW than the other groups. *L. reuteri* treatment, rather than butyrate treatment, improved HF-diet-related BW increases. The biochemical markers are in Appendix A.

For IPGTT, the HF group showed a higher glucose level at 15 min than the control group (HF group vs. control group; 233.33 ± 42.89 mg/dL vs. 386.20 ± 57.93 mg/dL; *p* = 0.021). Only *L. reuteri* treatment improved the IPGTT glucose level at 15 min (HF group vs. HF plus *L. reuteri* group; 386.20 ± 57.93 mg/dL vs. 231.44 ± 25.63 mg/dL; *p* = 0.022). (Appendix A). Neither *L. reuteri* nor butyrate treatment affected the area under the curve (AUC) level of IPGTT.

The HF diet increased caloric intake in the dams. *L. reuteri* treatment led to a decrease in both the total intake (Figure 2A) and intake per unit of BW (Figure 2B) in dams. This suggests that *L. reuteri* therapy may reduce the appetite of dams. A HF diet intake led to BP elevation, increased plasma leptin levels, and increased retroperitoneal fat deposits (Figure 2). Both *L. reuteri* and butyrate therapy improved the elevated BP related to the HF diet (Figure 2D). *L. reuteri* and butyrate treatments did not decrease plasma leptin levels or retroperitoneal fat deposits (Figure 2C,E). Thus, *L. reuteri* treatment during pregnancy ameliorates HF diet-related obesity through a leptin-independent pathway.

### 3.2. L. reuteri Is More Effective Than Butyrate in Preventing HF Diet-Induced Liver Steatosis in Pregnant Dams

HF diets may lead to liver steatosis and oxidative stress in pregnant dams [13]. We determined the preventive effects of *L. reuteri* and butyrate intake on HF diet-related fatty liver disease in pregnant dams. HF intake resulted in macrovesicular steatosis (a single lipid droplet filling the entire cell with a displaced nucleus) and microvesicular steatosis (several small cytoplasmic vacuoles with a central nucleus) (Figure 3A). *L. reuteri* treatment during pregnancy greatly improved macrovesicular and microvesicular steatosis in dams induced by a HF diet. Butyrate treatment during pregnancy primarily improved macrovesicular steatosis compared to microvesicular steatosis. Exposure to a HF diet also resulted in liver oxidative stress in dams, as illustrated by 8-OHdG staining, one of the major products of DNA oxidation. Both *L. reuteri* and butyrate therapies improved liver oxidative stress in the dams (Figure 3B).

### 3.3. Maternal L. reuteri Intake Was an Effective Preventative Strategy for Maternal HF Diet-Induced Fetal Liver Steatosis

Maternal HF diet intake also caused steatosis and oxidative stress damage in the fetal liver (Figure 4A,B). We evaluated the preventive effects of maternal *L. reuteri* and butyric acid intake during pregnancy on fetal liver steatosis. We found that, although both maternal *L. reuteri* and butyric acid could ameliorate oxidative stress injury in the fetal liver, only *L. reuteri* was effective in improving fetal fatty liver related to the maternal HF diet.

Since only maternal *L. reuteri* treatment could prevent hepatic steatosis in fetuses with prenatal HF exposure, we studied the mechanism by which prenatal *L. reuteri* intake could prevent fetal fatty liver. We found that acetyl-CoA carboxylase (ACC1) and lipoprotein lipase (LPL), coding for key enzymes implicated in lipid metabolism, were significantly increased in the maternal HF diet [13]. Prenatal *L. reuteri* treatment significantly improved the altered expressions of *ACC1* and *LPL* (Figure 5).

### 3.4. Maternal L. reuteri Treatment Can Reform Placenta Remodeling and Decrease Placental Oxidative Injury Induced by Maternal HF Diet

The mechanisms by which HF diet intake during pregnancy leads to obesity in the offspring include remodeling of the placenta and altered maternal gut microbiota composition. These mechanisms result in perturbed lipid metabolism in the fetal liver [13]. Placental remodeling and maternal gut microbiome shaping after *L. reuteri* treatment during maternal pregnancy were investigated. Placental histology showed that *L. reuteri* intake during pregnancy reformed placental remodeling related to the maternal HF diet. *L. reuteri* treatment during pregnancy increased the labyrinth zone and decreased the transitional zone compared to those in the HF group (Figure 6A). *L. reuteri* treatment during pregnancy diminished placental oxidative injury caused by maternal HF diet exposure (Figure 6B).

The placenta bridges maternal and fetal circulation. The placenta plays a key role in fetal development, inclusive of nutrient transportation, hormone production, and acting as an immune barrier [33]. In response to changes in maternal nutrition, the placenta undergoes functional and structural changes that affect the supply of nutrients, hormones, and other molecules to the fetus. Maintaining a balance between caloric intake and energy storage is important for ensuring good health. Cells rely on nutrient sensing to recognize and respond to fuels. Thus, nutrient sensing plays a key role in regulating tissue growth. Gene expressions of several important nutrient-sensing molecules in placental tissue were investigated. *L. reuteri* treatment during pregnancy broadly increased the mRNA expression of nutrient-sensing molecules including Sirtuin 1 (*SIRT-1*), peroxisome proliferator-activated receptor α (*PPAR-γ*), mammalian target of rapamycin (*mTOR*), and peroxisome proliferator-activated receptor gamma coactivator 1α (*PGC1α* ) (Figure 7).

### 3.5. Maternal L. reuteri Therapy Modified the SCFAs Signaling Cascade

Dietary manipulation of the host can lead to significant changes in the gut microbial composition and subsequent alteration of microbial metabolites. Therefore, we investigated the ability of *L. reuteri* treatment during pregnancy to alter the maternal gut microbiome or metabolites related to a HF diet. The HF diet for dams led to lower α-diversity and higher β-diversity of the gut microbiome in comparison to the CC diet (Figure 8A,B). This result suggested that taxa abundance was influenced by the HF diet. *L. reuteri* treatment did not alter the abundance of lower taxa related to the HF diet. Neither HF diet exposure nor *L. reuteri* treatment changed the F/B ratio (Figure 8C). In a previous study, we showed that an HF diet caused gut dysbiosis in dams [13]. In this study, we aimed to determine whether *L. reuteri* treatment during pregnancy can ameliorate dysbiosis associated with an HF diet. Among the taxa with the greatest difference between the CC and HF groups, the top ten most abundant and least abundant taxa in the CC group compared to the HF group were presented in the heatmap (Figure 8D). The corresponding taxa of the L group and H + L groups were also presented. In agreement with our previous findings, there was a higher presence of the *Akkermansia* genus in the gut microbiota of HF dams and less *Lachospiraceae* genus compared to CC dams [13]. In this study, HF dams also demonstrated more *Tannerellaceae* and *Parabacteroides* and lower *Turicibacter* genera than the CC group. The gut microbiota of the H + L group was similar to that of the HF group (Figure 8D). Since *L. reuteri* treatment could improve liver steatosis in dams and offspring related to the maternal HF diet, further work was carried out to identify the characteristic bacteria between the maternal HF diet with and without *L. reuteri* treatment. The tax abundance of HF dams with and without *L. reuteri* treatment was investigated using linear discriminant analysis (LDA) and the effect size (LEfSe) analysis (Figure 8D,E). The length of the histogram (i.e., LDA score), which was based on the size of the different species, was set to be significant, with a difference of more than 3 log10. At the genus level, the gut microbiomes of the HF and H + L groups showed similar performances. LEfSe also showed an increase only in *L. reuteri* in the H + L group compared with that in the HF group.

Considering the significant effect of probiotic treatment on metabolic modulation in offspring, probiotic treatment is likely to achieve therapeutic effects through relevant metabolites. SCFAs are important metabolites fermented from dietary fiber by the gut microbiota. G-protein-coupled receptors (GPR)41 and GPR43 are important SCFA receptors that respond to acetate, butyrate, and propionate. In the placental analysis, we found that probiotic treatment improved the corresponding mRNA expression of *GPR43* relative to maternal HF diet exposure (Figure 9B), whereas the mRNA expression of *GPR41* and *GPR120* was unaffected (Figure 9A,C). Our results also revealed that the plasma acetate and butyrate levels were similar across the four groups (Figure 9D,F). Probiotic treatment improved low plasma propionate outcomes following exposure to a maternal HF diet (Figure 9E).

## 4. Discussion

In this study, we tested the therapeutic effects of *L. reuteri* (a probiotic) and butyrate (a postbiotic) on metabolic problems during pregnancy in high-fat-fed obese dams. We found that *L. reuteri* intervention during pregnancy ameliorated obesity and altered the metabolic profiles of obese dams. Both *L. reuteri* and butyrate treatments improved liver steatosis related to the HF diet in pregnant dams; however, *L. reuteri* was more effective than butyrate. It was found that *L. reuteri* intake rather than butyrate intake was an effective preventative strategy for maternal HF diet-induced fetal liver steatosis. Maternal *L. reuteri* treatment was found to have the ability to reform placental remodeling and decrease placental oxidative injury induced by maternal HF diet. Maternal *L. reuteri* therapy also modifies the SFCAs signaling cascade, transcription of nutrient-sensing molecules in the placenta, and lipid metabolism in the fetal liver (Appendix A).

The diagnostic criteria for NAFLD are equal to or more than 5% liver fat accumulation and exclusion of other secondary hepatic fat accumulation causes, such as autoimmune liver disease, viral hepatitis, and significant alcohol intake [34]. NAFLD can be subdivided into steatosis (increased liver fat without inflammation) and steatohepatitis (increased liver fat with inflammation and hepatocellular injury). Hepatic steatosis and steatohepatitis may mirror different disease entities [35]. Under certain conditions, inflammation may precede steatosis in steatohepatitis. Whilst under other conditions, some patients with simple steatosis may develop steatohepatitis. Tilg et al. proposed the “multiple parallel hits” hypothesis to explain these diverse courses [35]. They include endoplasmic reticulum stress, endotoxins, energy metabolism changes by gut microbiota, aryl hydrocarbon receptor activation by trans-fatty acids and fructose, and an imbalance of adipocytokines. In our study, HF diet exposure in pregnant dams led to hepatic steatosis in dams and fetuses, rather than steatohepatitis. This may be due to the difference in hits. In addition, the development of steatohepatitis would likely have taken a longer time and was therefore not visible in the fetal period.

The gut microbiome plays an important role in the development of NAFLD. Studies have shown (in comparison to control animals) that received fecal transplants from insulin-sensitive mice, germ-free mice transplanted with gut microbiota from insulin-resistant mice that received a HF diet showed similar body weight and food consumption, but increased liver fat deposition [36]. The expression of key transcription factors that regulate lipogenesis was also increased in the liver. Gut bacteria also influence the host metabolism through their metabolites. One example of this is SCFAs. SCFAs are one of the end products of polysaccharides after fermentation by intestinal bacteria and they play a role in nutrient sensing and energy balance [37]. In a previous study, we found that pregnant dams fed a HF diet showed gut dysbiosis and a decrease in plasma propionate levels [13]. In this study, the intestinal flora of *L. reuteri*-treated pregnant dams exposed to the HF diet did not differ from those fed the HF diet alone. The gut microbiome of the *L. reuteri*-treated group showed a high abundance of *L. reuter**i*; this can be explained by the probiotic treatment. However, *L. reuteri* treatment increased the serum butyrate levels of pregnant dams and the gene expression of *GPR43* in placental tissue. A previous report showed that New Zealand black × New Zealand white (NZB/W) F1 mice supplemented with *L. reuteri* GMNL-89 had improved hepatic apoptosis and inflammatory markers [38]. In this study, *L. reuteri* treatment of pregnant dams showed dual effects of treating NAFLD in pregnant women with an HF diet and re-programming the fatty liver change in the fetus. These therapeutic effects may be partly mediated by the modulation of postbiotics.

To carry out their health effects, probiotics must overcome inappropriate lethal conditions during processing, and maintain their survival under inappropriate storage and digestive system conditions. Postbiotics may successfully overcome these adverse effects and become ideal substitutes for probiotics. In this study, we assigned butyrate as a postbiotic for the treatment of pregnant dams fed a HF diet. Previous studies showed that butyrate treatment relieves HF diet-induced hepatic steatosis in mice by upregulating miR-150 expression, downregulating C-X-C chemokine receptor type 4, enhancing insulin-induced gene activity, and suppressing lipogenic genes [25,39]. Modulation of intestinal tight junctions is also involved in the regulation of lipid metabolism in db/db mice [40]. In our study, we observed the beneficial effect of butyrate on hepatic steatosis in dams, even though *L. reuteri* treatment was more effective. Butyrate treatment did not have a therapeutic effect on fetal steatosis in mothers with obesity. Postbiotics are bioactive metabolites produced by gut bacteria during fermentation or may be structural fragments of bacteria. In addition to SCFAs, teichoic acids, polysaccharides, vitamins, peptides, plasmalogens, and organic acids are also important postbiotics with different functional properties [41]. Butyrate treatment alone cannot completely replace probiotics, and it is important to recognize the characteristics and functions of each postbiotic.

Pregnant obese women may pass on their metabolic phenotypes to their offspring, leading to a cycle of obesity across generations. Offspring of obese women are at an increased risk of developing NAFLD. A nationwide study in Sweden that included 165 young adults aged <25 years with liver biopsy-proven fatty liver disease and 717 age- and sex-matched non-fatty liver controls found that the offspring of obese mothers had a higher risk of developing fatty liver disease [42]. The increase in liver fat in the offspring of these obese mothers may be partially due to the delivery of more nutrients to the fetus. Fetal subcutaneous fat storage capacity is limited, so, the fetus is forced to use the liver to store fat. This results in a fatty liver [43]. The persistence of hepatic fat storage increase in the offspring of obese mothers may be due to changes in hepatic lipogenesis, fatty acid oxidation, or lipoprotein output process [13,43,44].

It has been shown that the expression of nutrient-sensing mitochondrial proteins SIRT1 and SIRT3 is reduced in the livers of offspring born to dams exposed to a HF diet and obese dams, respectively [45,46]. SIRT1 can modulate the transcription of peroxisome proliferator-activated receptor α (*PPARα*) and *PPARγ*, which in turn control fatty acid oxidation and lipogenesis, respectively. This indicates that fatty liver in the offspring of obese mothers is associated with nutrient-sensing defects and mitochondrial dysfunction. From our study, maternal *L. reuteri* treatment (rather than maternal HF diet alone) enhanced the mRNA expressions of several nutrient sensing molecules. The modulatory effects of nutrient-sensing molecules could be another therapeutic mechanism for prenatal *L. reuteri* treatment.

Our study has several limitations. Ideally, the expression of target genes should be evaluated at both the transcriptional and translational levels. Limited sample sizes inhibited the ability to analyze both transcriptional and translational levels in this study. As such, only mRNA expression, rather than protein abundance, was provided in this study. Further studies are needed to validate our observations at both the protein and functional levels. In our study, the α-diversity and β-diversity (Figure 8A,B) of the gut microbiome in the HF and H + L groups showed large deviations. In addition, the gut microbiomes of the HF and H + L groups were found to be significantly discrete (Figure 8C). Since food and sterile tap water were provided ad libitum in this study, it is possible that the gut microbiome was affected by the inconsistent eating status of each mouse. Therefore, the gut microbiome exhibits intragroup variation.

In this study, we found that pregnant dams exposed to a HF diet with *L. reuteri* had greater reductions in hepatic fat compared to dams fed the HF diet alone. It is possible that this effect was partially mediated by weight loss since the dams with *L. reuteri* intake lost body weight compared to those consuming only the HF diet. Because the caloric intake per body weight of dams fed *L. reuteri* was lower than that of dams not fed *L. reuteri*, *L. reuteri* consumption appears to reduce the appetite of dams fed an HF diet. Pharmacological treatments for NASH include aldafermin, pioglitazone, vitamin E, obeticholic acid, resveratrol, liraglutide, and lanifibranor [47,48]. *L. reuteri* has the potential to be used as adjuvant therapy for lifestyle interventions in the treatment of NAFLD in pregnant mothers and as a de-programming strategy for NAFLD in the next generation. The therapeutic effects of *L. reuteri* should be validated in future clinical trials.

## Figures and Tables

**Figure 1 nutrients-14-04004-f001:**
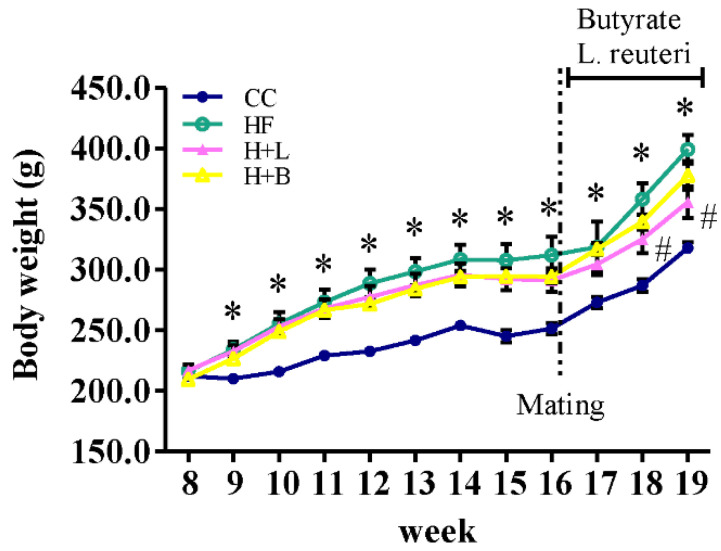
The change in body weight (BW) of dams with the control diet (CC), high-fat diet (HF), maternal high-fat diet with *Lactobacillus reuteri* treatment after mating (H + L), and maternal high-fat diet with sodium butyrate treatment after mating (H + B). A significant difference was observed after 1 week of diet manipulation. * Compared to the CC group, *p* < 0.05; # compared to the HF group, *p* < 0.05. The detailed data are shown in Appendix A.

**Figure 2 nutrients-14-04004-f002:**
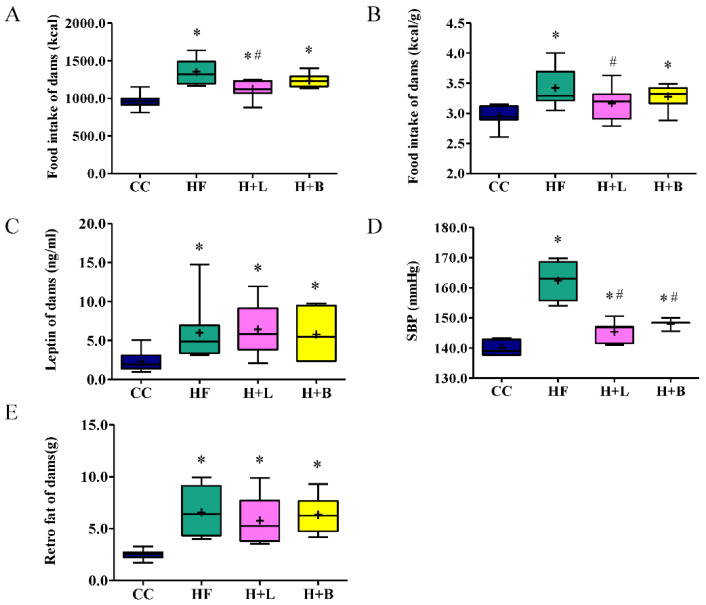
Metabolic profiles of the dams before delivery. (**A**) The total calorie intake of dams during the study period; (**B**) caloric intake per unit of BW; (**C**) plasma leptin level of dams; (**D**) systolic blood pressure of dams before delivery; (**E**) weight of retroperitoneal fat deposits. Data are shown as a box and whisker plot. The box is bounded on the top by the third quartile, and on the bottom by the first quartile. The median divides the box. The diagram also shows the mean level (cross symbol) “+”. The whiskers are the two lines outside the box. One extends upward from the third quartile (Q3) to the maximum, and the other goes downward from the first quartile (Q1) to the minimum. Data falling outside the Q1 to Q3 range are plotted as outliers of the data. * Compared to the CC group, *p* < 0.05; # compared to the HF group, *p* < 0.05. SBP, systolic blood pressure; retro, retroperitoneal.

**Figure 3 nutrients-14-04004-f003:**
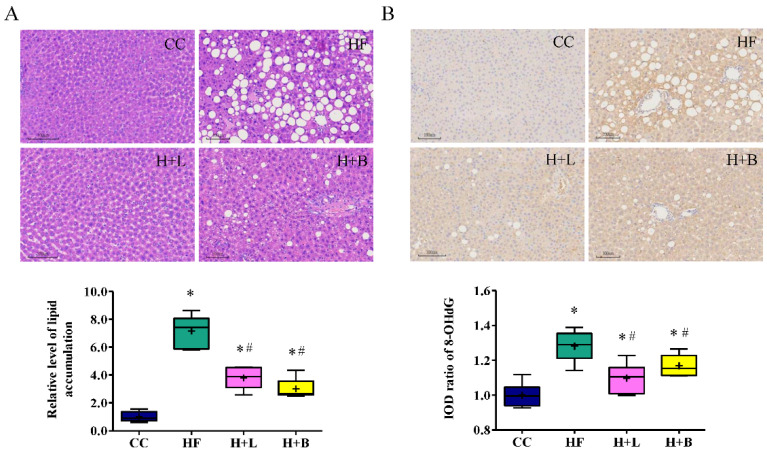
Histological manifestations of dam livers with high-fat diets and *Lactobacillus reuteri* or sodium butyrate treatment. (**A**) The degree of hepatic steatosis increased with a high-fat diet (HF) and presented as a vacuolation increase via hematoxylin and eosin (H&E) stain. High-fat diet intake caused obvious macrovesicular steatosis (a single lipid droplet filled the entire cell with the displaced nucleus) and microvesicular steatosis (several small cytoplasmic vacuoles with a centrally placed nucleus). *L. reuteri* treatment during pregnancy (H + L) substantially improved hepatic macrovesicular steatosis and microvesicular steatosis of dams. Sodium butyrate treatment during pregnancy (H + B) mainly improved macrovesicular steatosis and had less effect on microvesicular steatosis. (**B**) Hepatic oxidative stress of dams determined by 8-hydroxy-2-deoxyguanosine (8-OHdG) staining. The liver oxidative stress of dams is highest in the HF group. Both *L. reuteri* and butyrate therapies during pregnancy improved hepatic oxidative stress related to a high-fat diet. * Compared to the control diet (CC) group, *p* < 0.05; # compared to the HF group, *p* < 0.05. IOD, image optical density. The diagram shows the mean level (cross symbol) “+”.

**Figure 4 nutrients-14-04004-f004:**
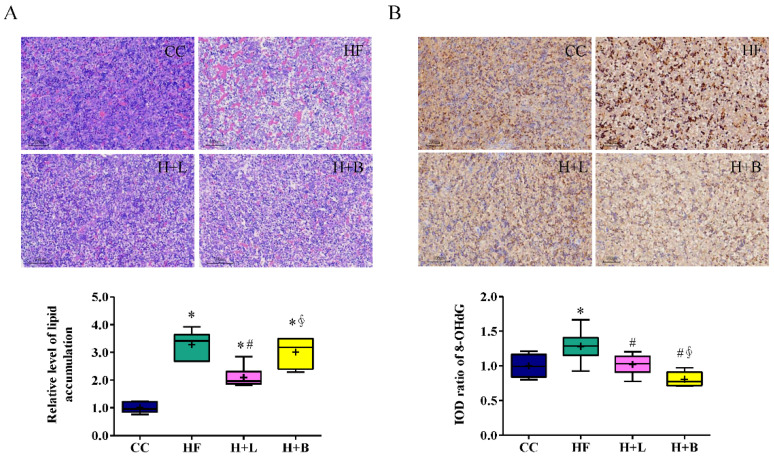
Histological manifestations of the fetal liver with a maternal high-fat diet (HF) and prenatal *Lactobacillus reuteri* (H + L) or prenatal sodium butyrate treatment (H + B). (**A**) The degree of fetal hepatic steatosis is presented as vacuolation in the H&E stain. (**B**) The fetal liver oxidative stress was higher with a maternal high-fat diet (HF) compared to a controlled maternal diet (CC); 8-hydroxy-2-deoxyguanosine (8-OHdG) was used to determine oxidative stress. * Compared to the CC group, *p* < 0.05; # compared to the HF group, *p* < 0.05; ∮ compared to the H + L group, *p* < 0.05. IOD, image optical density. The diagram shows the mean level (cross symbol) “+”.

**Figure 5 nutrients-14-04004-f005:**
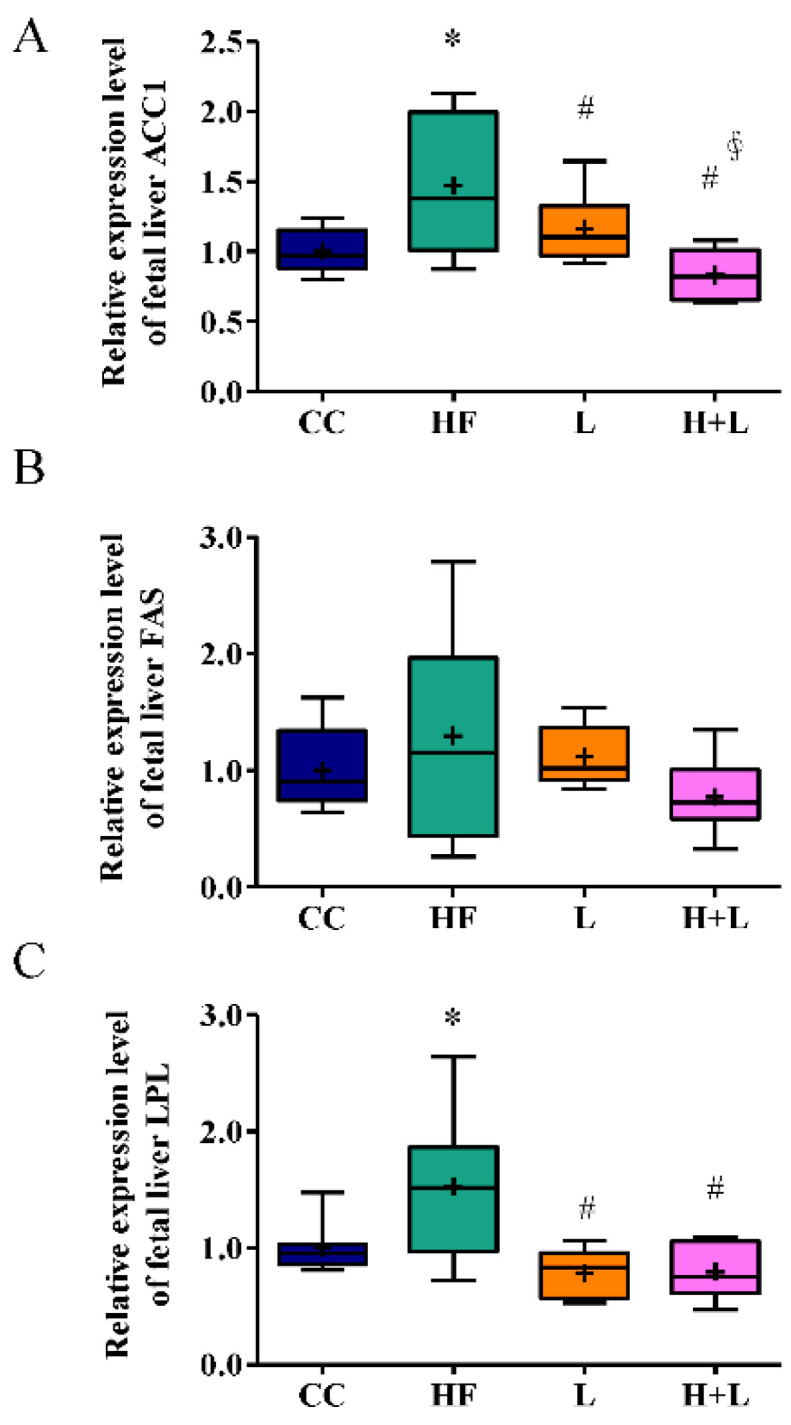
Gene expressions of key enzymes for lipid metabolism in fetal liver. (**A**) ACC1 (**B**) FAS (**C**) LPL mRNA expressions in the fetal liver with a maternal control diet (CC), a maternal high-fat diet (HF), maternal control diet with prenatal Lactobacillus reuteri (L) treatment, or a maternal high-fat diet with prenatal Lactobacillus reuteri treatment (H + L). The housekeeping gene was GAPDH and mRNA expression is shown in the relative fold. * Compared to the CC group, *p* < 0.05; # compared to the HF group, *p* < 0.05; ∮ compared to the L group, *p* < 0.05. ACC1, acetyl-CoA carboxylase 1; FAS, fatty acid synthases; LPL, lipoprotein lipase; GAPDH, glyceraldehyde-3-phosphate dehydrogenase. The diagram shows the mean level (cross symbol) “+”.

**Figure 6 nutrients-14-04004-f006:**
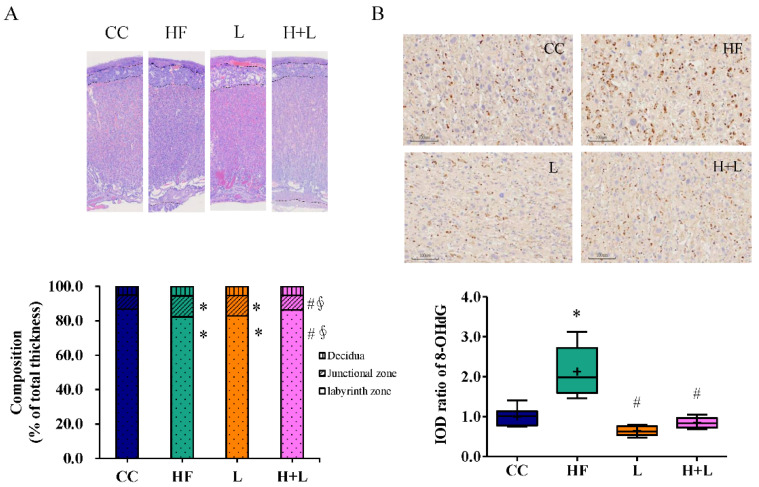
Histological changes in placenta with maternal high-fat diet and *Lactobacillus reuteri* therapy (**A**) Histological manifestations of the placenta. The mean proportions of the thickness of the decidua basalis, junctional zone, and labyrinth zone are shown. (**B**) Oxidative stress was determined by 8-hydroxy-2-deoxyguanosine (8-OHdG) staining. * Compared to the CC group, *p* < 0.05; # compared to the HF group, *p* < 0.05; ∮ compared to the L group, *p* < 0.05. CC, maternal control diet group; HF, maternal high-fat diet group; L, maternal control diet with prenatal *Lactobacillus reuteri* treatment; H + L, maternal high-fat diet with prenatal *Lactobacillus reuteri* treatment; IOD, image optical density. The diagram shows the mean level (cross symbol) “+”.

**Figure 7 nutrients-14-04004-f007:**
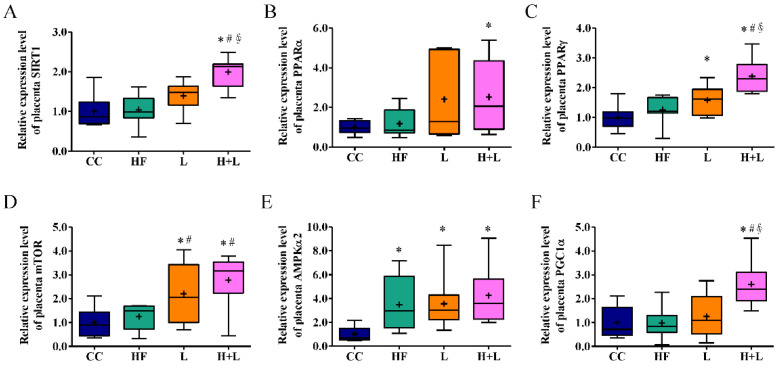
Gene expressions of nutrient sensing molecules of placenta tissue. (**A***) SIRT-1* (**B**) *PPARα* (**C**) *PPARγ* (**D**) *mTOR* (**E**) *AMPKα2* (**F**) *PGC1α* mRNA expressions in placenta tissue with maternal control diet (CC), maternal high-fat diet (HF), maternal control diet with prenatal *Lactobacillus reuteri* (L) treatment, or maternal high-fat diet with prenatal *Lactobacillus* reuteri treatment (H + L). The mRNA expression is shown in the relative fold. * compared to the CC group, *p* < 0.05; # compared to the HF group, *p* < 0.05; ∮ compared to the L group, *p* < 0.05. SIRT1, Sirtuin 1; *PPARα*, Peroxisome proliferator-activated receptor α; *PPARγ*, peroxisome proliferator-activated receptor γ; *mTOR*, mammalian target of rapamycin; *AMPKα2*, AMP-activated protein kinase; *PGC1**α*, peroxisome proliferator-activated receptor gamma coactivator 1α. The diagram shows the mean level (cross symbol) “+”.

**Figure 8 nutrients-14-04004-f008:**
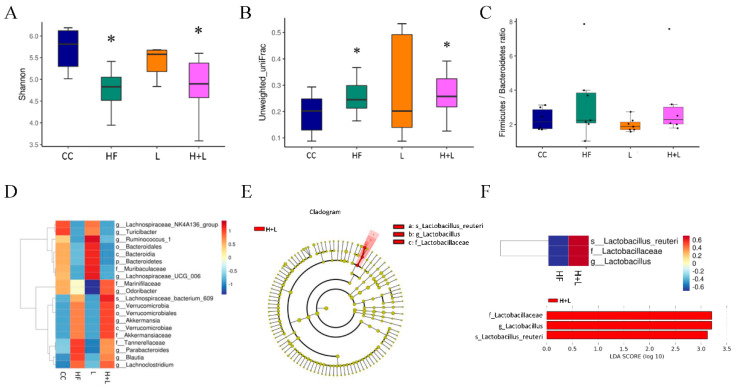
The effects of a high-fat diet and *L. reuteri* treatment during pregnancy on the gut microbiome of dams. (**A**) α-diversity (**B**) β-diversity (**C**) *Firmicutes/Bacteroidetes* (F/B) ratio (**D**) Heat map showing the relative abundance of microbiomes among the four groups. (**E**) Cladogram indicated that *L. reuteri* was the predominant bacteria in the HF and H + L groups. (**F**) For the linear discriminant analysis (LDA) distribution histogram, taxa that reached a linear discriminant analysis score (log10) >3.0 are highlighted and labeled. LDA and effect size (LEfSe) also showed that only *L. reuteri* increased in the H + L group compared to the HF group. * compared to the CC group, *p* < 0.05. CC, maternal control diet group; HF, maternal high-fat diet group; L, maternal control diet with prenatal *L. reuteri* treatment; H + L, maternal high-fat diet with prenatal *L. reuteri* treatment.

**Figure 9 nutrients-14-04004-f009:**
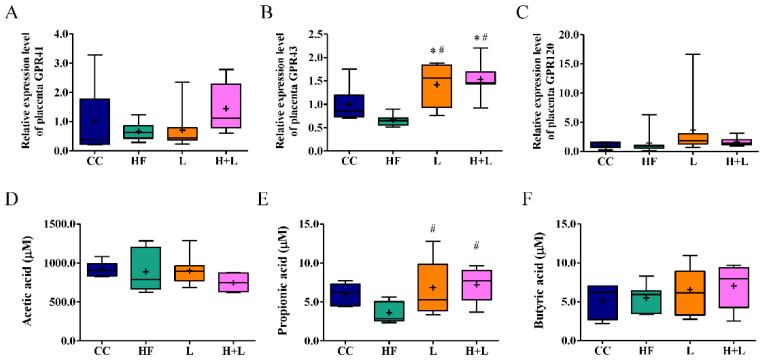
The effects of *Lactobacillus reuteri* treatment on the short-chain fatty acid cascade. mRNA expression of (**A**) *GPR41* (**B**) *GPR43* (**C**) *GPR120* in placental tissue. The housekeeping gene was 18S rRNA. mRNA expression is presented in terms of relative fold. Plasma (**D**) acetic acid, (**E**) propionic acid, and (**F**) butyric acid levels of dams as determined by gas chromatography. * Compared to the CC group, *p* < 0.05; # compared to the HF group, *p* < 0.05. CC, maternal control diet group; HF, maternal high-fat diet group; L, maternal control diet with prenatal *L. reuteri* treatment; H + L, maternal high-fat diet with prenatal *L. reuteri* treatment; GPR, G-protein-coupled receptor. The diagram shows the mean level (cross symbol) “+”.

## Data Availability

Data are available from the corresponding author upon publication.

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
