# Peer review of "Effects of Maternal Gut Microbiota-Targeted Therapy on the Programming of Nonalcoholic Fatty Liver Disease in Dams and Fetuses, Related to a Prenatal High-Fat Diet"

_nutrients, 2022, doi:10.3390/nu14194004_

Round 1
Reviewer 1 Report
This study investigated whether perinatal high-fat diet-induced programmed hepatic steatosis in fetal offspring can be prevented by the therapeutic use of probiotic Lactobacillus reuteri or by postbiotic butyrate gestation. Overall, the study is interesting novel findings and provides well-executed experiments with appropriate controls, I am therefore very supportive of this study. However, the paper should be major improved. I have listed some comments below:
1. Section 2.1 and 2.5: Estimation of the sample size and power is highly suggested. The sample size in each group was 6, not large, with 4 groups. Because the study is essentially an exploratory analysis, not a hypothesis-driven one, the relatively small sample size may be acceptable from my perspective. However, the authors need to indicate an estimate of beta-level (power) based on the parameters used.
2. Section 2.1: Please provide the animal ethics number.
3. Section 2.1: ad libitum should be in italics.
4. Section 2.1: It is suggested to add an experimental design figure.
5. Section 2.2: Please supplement the whole name of RT-qPCR.
6. Section 2.2: Please provide the method of killing the rats. How much blood was collected and what was the technique used? The vein or cardiac bleed? Cheek bleed or retro-orbital bleed?
7. Section 2.7: Why GAPDH and 18S ribosomal RNA were chosen as housekeeping genes in two different tissues. In this paper, the authors explore the relationship between NAFLD and oxidative stress. In the case of body oxidation, NADPH will be regulated and changed, I think it is not suitable as a housekeeping gene here, instead, β-actin may be better.
8. Section 2.8: Please indicate whether Dada2 or Deblur was used in the first QC step of the Qiime2 analysis pipeline. Please indicate how the data were rarefied for microbiome diversity analysis. It is important to rarefy for diversity analyses to avoid a depth sampling bias.
9. Uniform P and the name of Lactobacillus reuteri are italic or non-italic throughout the whole article.
10. Why didn’t the author test for biochemical markers of the liver besides slices?
11. References should be updated also evaluating the contribution of more recent studies available on the area of interest.
12. It is hard to distinguish scale in the figure 4.
13. What is the meaning of “+” in all figures?
14. Please use the same font size for the “*” in figure 8
15. The gene expression of target genes should be measured at both transcriptional (mRNA) and translational (protein, Western blot analysis) levels. Without having such data, the conclusion cannot be finally drawn.
16. Figure 8D, please provide the selection method of the genus in figure 8D, top10, or other standard.
Reviewer 2 Report
In this study, the authors aimed to investigate Lactobacillus reuteri and sodium butyrate treatment in an attempt to attenuate the effects of maternal obesity on offspring in pregnant dams. This study is novel and of great interest. However, there are several points that still need improvement.
1. Please delete the other extra information after the “kewords” section. Also, please confirm the correct expression of “prenatal high fat diet” in the “kewords” section and keep it consistent throughout the manuscript.
2. It is recommended to use a table to compare the control chow diet and HF diet with the feed ingredients. This is a clearer way to present.
Please specify the specific grouping. For example, how does “H+F”、“L+B” stand for grouping? Like the expression“Group I CC”.
3. Please provide references support for the method of “Intraperitoneal Glucose Tolerance Test”.
4. Please briefly describe the specific method of the “automatic biochemical analyzer”and “enzyme-linked immunosorbent assay kit” to facilitate understanding. Support with references if necessary.
5. Please adjust the format of figure 5.
6. Please revise the statement on short-chain fatty acids in the results section "3.5"?
7. In the figure 8A and figure 8B, the histograms of L and H+F groups show a large deviation. In addition, the samples of HF and H+F groups are found to be significantly discrete in figure 8C. What puzzles me is what causes this phenomenon? Are such results a strong indication of the conclusion?
8. In the notes of figure 8F, only the meaning of the LDA diagram is elaborated, not the heatmap. Moreover, I do not really understand the significance of the authors' use of heatmap? It is well known that the LEfSe analysis is a screening of different groups of characteristic bacteria. In the figure 8F the LDA diagram only screens for the characteristic bacteria in the H+F group, so why should it be compared with the HF group? What's more, the characteristic bacteria in the HF group may be different from the H+F group.
9. Please use abbreviations for some of the expressions in the “discussion” section.
10. The language used in the manuscript is rather homogeneous and it is recommended that the quality of the manuscript writing be improved.
Round 2
Reviewer 1 Report
The manuscript is well revised.
Reviewer 2 Report
I have no more comment.